# Augmented Ouabain-Induced Vascular Response Reduces Cardiac Efficiency in Mice with Migraine-Associated Mutation in the Na^+^, K^+^-ATPase α_2_-Isoform

**DOI:** 10.3390/biomedicines11020344

**Published:** 2023-01-25

**Authors:** Rajkumar Rajanathan, Tina Myhre Pedersen, Halvor Osterby Guldbrandsen, Lenette Foldager Olesen, Morten B. Thomsen, Hans Erik Bøtker, Vladimir V. Matchkov

**Affiliations:** 1Department of Biomedicine, Aarhus University, 8000 Aarhus, Denmark; 2Department of Biomedical Sciences, University of Copenhagen, 1165 Copenhagen, Denmark; 3Department of Cardiology, Aarhus University Hospital, 8000 Aarhus, Denmark

**Keywords:** ouabain, hemodynamic, migraine, cardiovascular function, in vivo, cardiac pacing, total peripheral resistance, Na^+^,K^+^-ATPase, α_2_-isoform

## Abstract

Heterozygous mice (α_2_^+/G301R^ mice) for the migraine-associated mutation (G301R) in the Na^+^,K^+^-ATPase α_2_-isoform have decreased expression of cardiovascular α_2_-isoform. The α_2_^+/G301R^ mice exhibit a pro-contractile vascular phenotype associated with decreased left ventricular ejection fraction. However, the integrated functional cardiovascular consequences of this phenotype remain to be addressed in vivo. We hypothesized that the vascular response to α_2_-isoform-specific inhibition of the Na^+^,K^+^-ATPase by ouabain is augmented in α_2_^+/G301R^ mice leading to reduced cardiac efficiency. Thus, we aimed to assess the functional contribution of the α_2_-isoform to in vivo cardiovascular function of wild-type (WT) and α_2_^+/G301R^ mice. Blood pressure, stroke volume, heart rate, total peripheral resistance, arterial dP/dt, and systolic time intervals were assessed in anesthetized WT and α_2_^+/G301R^ mice. To address rate-dependent cardiac changes, cardiovascular variables were compared before and after intraperitoneal injection of ouabain (1.5 mg/kg) or vehicle during atrial pacing. The α_2_^+/G301R^ mice showed an enhanced ouabain-induced increase in total peripheral resistance associated with reduced efficiency of systolic development compared to WT. When the hearts were paced, ouabain reduced stroke volume in α_2_^+/G301R^ mice. In conclusion, the ouabain-induced vascular response was augmented in α_2_^+/G301R^ mice with consequent suppression of cardiac function.

## 1. Introduction

Since its discovery in 1957, the Na^+^,K^+^-ATPase has been extensively studied [1,2,3]. The Na^+^,K^+^-ATPase transports three Na^+^ ions across the cell membrane to the extracellular milieu, in return, two K^+^ ions are transported into the cell [4]. This electrogenic ion transport is crucial for many cellular functions including stabilization of the membrane potential, control of cell volume, and maintaining electrochemical gradients important for the secondary active transport of other ions and substrates across the cell membrane [4]. Recently, the Na^+^,K^+^-ATPase has also been described to be involved in intracellular signaling [5,6]. The importance of the Na^+^,K^+^-ATPase is evident by its ubiquitous expression in all eucaryotic cells [7].

The Na^+^,K^+^-ATPase is a heterodimer transporter that consists of at least an α and a β subunit, and it can also include other regulatory and accessory subunits, e.g., FXYD proteins [8]. The α-subunit is highly conserved across species, and it is responsible for the catalytic and ion-transporting functions of the Na^+^,K^+^-ATPase [8].

So far, four Na^+^,K^+^-ATPase α-isoforms have been identified, i.e., the α_1_-, α_2_-, α_3_-, and α_4_-isoforms [2,3]. The α-β dimer containing the α_1_-isoform is commonly expressed in nearly all cell membranes and is generally thought to have a housekeeping role in controlling transmembrane ion homeostasis [3]. The α_2_-isoform is co-expressed with the α_1_-isoform in several cell types including muscle tissues, e.g., cardiomyocytes, skeletal, and vascular smooth muscle cells [2,3,9,10,11].

Changes in the Na^+^,K^+^-ATPase function are implicated in different pathologies including cardiovascular diseases [2,3,9,10,11,12,13,14,15]. In cardiomyocytes and in vascular smooth muscle cells, the Na^+^,K^+^-ATPase is also important for intracellular Ca^2+^ control and, thus, cardiac and smooth muscle cell contractility [16]. Canonically, the Na^+^,K^+^-ATPase generates the Na^+^ gradient required for the Na^+^/Ca^2+^ exchanger to extrude Ca^2+^ [16]. However, the isoform-specificity of this Ca^2+^ regulation has been heavily debated. Moreover, the molecular mechanisms underlying the association between isoform-specific functions and cardiovascular pathology remain unclear [3,17,18].

Familial hemiplegic migraine type 2 (FHM2) is a severe form of inherited migraine with aura [19]. Because FHM2 is associated with mutations in the gene encoding the Na^+^,K^+^-ATPase α_2_-isoform [20], this specific disarray offers an avenue for investigation of the α_2_-isoform-specific function in the cardiovascular system. Thus, this study used mice heterozygous for an FHM2-associated mutation (G301R), i.e., α_2_^+/G301R^ mice [21]. The G301R mutation leads to decreased expression of the Na^+^,K^+^-ATPase α_2_-isoform in both vascular and heart tissues [9,10,11]. In contrast, the expression of the α_1_-isoform is increased in the hearts of α_2_^+/G301R^ mice [11]. This inverse relationship between the expressions of the α_1_- and α_2_-isoforms has also been reported for other mouse models [15,22]. The α_2_^+/G301R^ mouse model is characterized by a changed vascular phenotype [9,10] and an age-dependent (>8 months old) dilation of the ventricles with an associated reduced left ventricular ejection fraction [11].

Cardiac glycosides are a group of molecules that are synthesized by plants and some amphibians [23]. Chemically, they may be characterized by a lactone ring and a steroid ring together with a sugar moiety [23]. The primary mechanism underlying the pharmacodynamics of cardiac glycosides is thought to be their inhibition of the Na^+^,K^+^-ATPase [23,24,25,26]. Cardiac glycosides are known in the clinic for their inotropic and chronotropic properties [27]. However, in research, cardiac glycosides may be used as a tool to characterize the functional contribution of the Na^+^,K^+^-ATPase to physiological and pathophysiological functions [3,17]. Ouabain is the cardiac glycoside most often used in basic research, and in rodents, ouabain inhibits the isoforms of the Na^+^,K^+^-ATPase differently in a concentration-dependent manner [3,17,28]. At low concentrations (below 1–10 µM), ouabain inhibits the α_2_-isoform and has close to no effect on the α_1_-isoform [28]. The α_1_-isoform is only inhibited by concentrations that are 100-fold higher due to a genetic variant unique to rodents [8,28]. Consequently, ouabain allows distinction between isoform-specific functions in rodents. 

The vascular effects of ouabain are disputed [24,25,26,29]. While some studies indicate that long-term ouabain administration in rats increases blood pressure in vivo [24,29], other studies suggest no effect of ouabain on blood pressure [25,26]. Additionally, ex vivo studies suggest both ouabain-induced impairment and augmentation of vascular contractility depending on the vascular bed [24,26,30]. This matter is further complicated in vivo due to the inotropic effects of ouabain, which may also contribute to alterations in blood pressure [31,32]. Whether the conflicting data are due to variations in the inhibition of isoform-specific functions of the Na^+^,K^+^-ATPase remains unknown. The integrated cardiovascular assessment in vivo may be helpful in distinguishing between the cardiac and vascular components of the effect of ouabain and α_2_-isoform function. We have previously validated an experimental setup that allows for: (i) Comprehensive assessment of cardiac and vascular functions; (ii) control of heart rate through atrial pacing and examination of rate-dependent variables; and (iii) distinction between simultaneous changes in cardiac and vascular variables in vivo in the anesthetized mouse [33]. Therefore, in this study, we aimed to comprehensively characterize the cardiovascular effect of ouabain in aged (>8 months old) α_2_^+/G301R^ mice, which were previously shown to have reduced left ventricular ejection fraction [11], and to compare cardiac and vascular responses to ouabain in α_2_^+/G301R^ mice and matching wild-type mice (WT).

We hypothesized that the effect of ouabain on cardiac and vascular functions is greater in α_2_^+/G301R^ mice compared to WT at plasma ouabain concentrations inhibiting the α_2_-isoform. This is due to the reduced expression of the α_2_-isoform in the cardiovascular system of α_2_^+/G301R^ mice resulting in a more efficient ouabain-induced inhibition compared to that of the WT.

Our study demonstrates the importance of Na^+^,K^+^-ATPase α_2_-isoform for vascular function and links the ouabain-responsive vascular phenotype in α_2_^+/G301R^ mice to consequences for their cardiac function in vivo.

## 2. Materials and Methods

### 2.1. Experimental Animals

According to the cardiac phenotype previously described in aged α_2_^+/G301R^ mice [11], we used aged α_2_^+/G301R^ and WT mice with C57BL/6J background (>8 months old) in this study. Investigators were blinded regarding the genotype of animals. The α_2_^+/G301R^ mouse line was generated as previously described [21], and they were kept and bred at Aarhus University, Denmark. The homozygous genotype for the G301R mutation is lethal, therefore, only heterozygous mice were available for this study [34,35]. No sex differences have previously been observed in the cardiovascular phenotype of α_2_^+/G301R^ mice [9,10,11]. Thus, due to their larger body sizes, only male mice (*n* = 12 for α_2_^+/G301R^ and *n* = 12 for WT) were used in this study. Our approach yielded a higher surgical success rate ultimately reducing the number of animals used.

Mice were housed in rooms with controlled temperature (21.5 °C) and humidity (55%) with a 12:12 h light-dark cycle. Mice had ad libitum access to food and water. All animal experiments conformed to the guidelines from Directive 2010/63/EU of the European Parliament on the protection of animals used for scientific purposes. The experimental protocol was approved by the Animal Experiments Inspectorate of the Danish Ministry of Environment and Food and reported in accordance with the ARRIVE (Animal Research: Reporting in vivo Experiments) guidelines [36]. Mice were euthanized under deep anesthesia by cervical dislocation at the end of the study.

### 2.2. In Vivo Cardiovascular Study

The surgical preparation of mice and implantation of catheter, probe, and electrodes were done as previously described [33]. Mouse core temperature was kept at 37 ± 0.5 °C with a homeothermic blanket system (50-7222F, Harvard Apparatus, Holliston, MA, USA). Mice were mechanically ventilated with a MiniVent Ventilator (model 845, Harvard Apparatus, Holliston, MA, USA) under isoflurane anesthesia (induced at 3% and maintained at 2% in 100% O_2_ throughout the experiment). Ventilation was connected to a capnograph (Type 340, Harvard Apparatus, Holliston, MA, USA) relaying end-tidal CO_2_ (%_et_CO_2_). Tidal volume relied on body mass (10 µL/g) [37]. Respiration rate was adjusted according to %_et_CO_2_ (approximately 3.5%) [38], and it was then kept constant. For arterial blood pressure measurements, the tip of a 1.0 F solid-state catheter (SPR-1000, Millar, Houston, TX, USA) was introduced into the right common carotid artery and placed in the aortic arc. A transit-time flow probe (1.5 SL, Transonic, Ithaca, NY, USA) soaked in ultrasound gel (Kruuse, Langeskov, Denmark) was latched onto the ascending aorta. Flow measured by this flow probe was used as a proxy for cardiac output. Platinum bipolar electrodes positioned on the sinus node were used for electrical atrial pacing and were connected to a dual bioamplifier/stimulator unit (ADInstruments, Sydney, Australia). The electrical pulse width and the current used were 0.2 ms and 3 mA, respectively. Electrocardiogram (ECG) leads I and II were recorded with electrodes on the paws (MLA2505, ADInstruments, Sydney, Australia). During surgical preparation, mice were subject to intraperitoneal (i.p.) administration of pancuronium-bromide (0.4 mg/kg; P1918, Sigma, St. Louis, MO, USA) for the inhibition of respiratory reflexes and NaCl solution (1 mL, 9 mg/mL NaCl; Fresenius Kabi, Bad Homburg, Germany) to compensate any fluid loss.

Instrumentation of the flow probe and blood pressure catheter was successful in all 24 mice. However, the ECG recording of one WT mouse in the vehicle group was not interpretable. Thus, systolic time intervals were not calculated for this mouse.

Following stabilization of cardiovascular variables under the set respiratory parameters, the protocol was initiated (Figure 1). The protocol consisted of two rounds of atrial pacing with administration of either ouabain (1.5 mg/kg i.p., Sigma, St. Louis, MO, USA) or a corresponding volume of vehicle (9 mg/mL i.p., NaCl). First, to control heart rate, atrial pacing was done at 10 Hz with stepwise increments to 11.3 Hz (678 BPM) and then kept for 20 s as previously described [33]. Then, after one minute of sinus rhythm, i.e., heart rate without atrial pacing, ouabain or vehicle was administered i.p., and 5 min were given for cardiovascular variables to restabilize. Eight animals from each genotype were randomized to ouabain administration, and four animals from each genotype were randomized to vehicle administration. Another atrial pacing session was performed at 10 Hz with stepwise increments to 11.3 Hz, which was also kept for 20 s. Subsequently, the protocol was finalized with 6 min of sinus rhythm.

At the end of the experiment, which was approximately 15 min after i.p. injection, blood was collected from the left ventricle in heparinized Eppendorf tubes. The whole blood was centrifuged in a Biofuge centrifuge (Biofuge Fresco, Heraeus, Hanau, Germany) at 2000 RPM (g-force of approximately 310 m/s^2^) and 4 °C for 10 min to separate the blood components. Plasma was collected and snap-frozen for plasma ouabain analysis. This included 4 and 5 samples in the vehicle groups together with 7 and 13 samples in the ouabain groups for α_2_^+/G301R^ and WT mice, respectively.

### 2.3. Measurements of Plasma Ouabain Concentration

Plasma ouabain concentrations were measured with an ELISA kit (CEV857Ge, Cloud-Clone Corp., Houston, TX, USA). The assay was prepared according to the manufacturer’s instructions. To increase the detection range for the standard ouabain concentration curve, we utilized the linearity of the standard ouabain concentration curve in ouabain concentration range of 50,000, 16,666.7, 5555.6, 1851.9, 617.3, 205.7, and 68.60 pg/mL. Each sample measurement was duplicated. The experiment and analysis were performed in a blinded fashion. The signal optical density was measured in a microplate spectrophotometer (Powerwave 340, BioTek, Winooski, VT, USA).

### 2.4. Data Acquisition and Calculations

All data were recorded in LabChart Pro 8 (ADInstruments, Sydney, Australia). Hemodynamic and ventilation variables were acquired at a sampling rate of 1 kHz using a Powerlab unit (PowerLab 16/35, ADInstruments, Sydney, Australia). The mouse ECG lead I and II were recorded with a sampling rate of 4 kHz. Hemodynamic variables were averaged over one minute. During atrial pacing, variables were averaged over a period of 20 s. Blood flow was measured as the integral of the transit-time flow measurement [39], and cardiac output was calculated based on the measured time interval. Stroke volume was calculated from heart rate and cardiac output. Total peripheral resistance was calculated from mean arterial pressure and cardiac output [40]. As indices for contractility and efficiency of relaxation, the maximum positive first-time derivative (dP/dt_max_) [41] and the minimum negative first-time derivative (dP/dt_min_) [42] were calculated based on arterial blood pressure measurements, respectively. The systolic time intervals were calculated to estimate myocardial efficiency [43]. This was done based on the temporal relationship between the QRS complex from lead I of the mouse ECG and the opening and closure of the aortic valve based on the blood pressure waveform and blood flow trace (Figure 2). For calculating the systolic time intervals, a script developed in MatLab (MatLab R2020a, MathWorks, Natick, MA, USA) was used. The pre-ejection period was defined as the time between the start of the QRS complex and the opening of the aortic valve indicated by the initial peak of blood flow acceleration in the aorta [44,45]. Left ventricular ejection time was defined as the time between opening of the aortic valve until its closure indicated by the dicrotic notch on the blood pressure waveform [44,46]. The systolic time interval ratio was calculated by dividing pre-ejection period by left ventricular ejection time [43]. Systolic duration was defined as the sum of the pre-ejection period and left ventricular ejection time.

### 2.5. Statistical Analysis

All data are presented as mean values ± SEM. Also, pacing data are presented as mean values with individual paired data points. If data violated the assumption of normality, a non-parametric test was used for analysis as indicated. Baseline parameters and cardiovascular variables were compared with an unpaired *t*-test or a Mann–Whitney test. In vivo functional data were analyzed with multiple Wilcoxon tests or a two-way repeated measures ANOVA. Dunnett’s or Bonferroni’s multiple comparison tests were used when a statistically significant effect of a variable was detected. Plasma ouabain concentrations were analyzed with a two-way ANOVA followed by Bonferroni’s multiple comparison test. *p*-values below 0.05 were considered statistically significant.

## 3. Results

### 3.1. Baseline Physiological Variables Were Similar between WT and α_2_^+/G301R^ Mice

All mice were of the same age and displayed similar body mass. The physiological variables assessed before atrial pacing and pharmacological intervention were also similar between anesthetized WT and α_2_^+/G301R^ mice (Table 1).

### 3.2. Administration of Ouabain Similarly Elevates the Plasma Ouabain Concentrations in WT and α_2_^+/G301R^ Mice

Injection of ouabain (1.5 mg/kg, i.p.,) significantly increased plasma ouabain concentrations of both WT and α_2_^+/G301R^ mice (*p* < 0.0001; Figure 3). There was no genotype effect in plasma ouabain concentrations before and after interventions (Figure 3).

### 3.3. Ouabain Elevated Peripheral Vascular Resistance and Blood Pressure While Cardiac Variables Were Unchanged in Both WT and α_2_^+/G301R^ Mice

In general, ouabain administration increased systolic blood pressure (*p* = 0.0193); however, following post-hoc analysis, the changes in WT systolic pressure did not achieve statistical significance (*p* = 0.0993, *p* = 0.0901, *p* = 0.0556, and *p* = 0.0944 for 2, 5, 9, and 13 min, respectively, after ouabain administration compared to the time prior injection; Figure 4a). The diastolic blood pressure was increased in both genotypes (*p* = 0.0100; Figure 4a). The increase in blood pressure was associated with ouabain-induced elevation in total peripheral resistance (*p* < 0.0001). The dynamic of these changes was steeper in α_2_^+/G301R^ mice than in WT mice (*p* = 0.0340; Figure 4b). Ouabain administration did not modify stroke volume, heart rate, or cardiac output (Figure 4c–e).

Ouabain administration significantly increased the systolic time interval ratio of α_2_^+/G301R^ mice only (*p* = 0.0004; Figure 5a). This was associated with ouabain-induced prolongation of the pre-ejection period (*p* = 0.0007) and shortening of the left ventricular ejection time (*p* = 0.0426; Figure 5b,c). Systolic time intervals of WT mice were unchanged. The systolic duration, arterial dP/dt_max_, and arterial dP/dt_min_ were not affected by ouabain administration in both α_2_^+/G301R^ and WT mice (Figure 5d–f).

Vehicle administration had no effect on blood pressure, total peripheral resistance, stroke volume, heart rate, nor cardiac output in both α_2_^+/G301R^ and WT mice (Appendix A). Similarly, no effect of vehicle administration was observed in systolic time intervals nor in systolic duration, arterial dP/dt_max_, and arterial dP/dt_min_ (Appendix A).

### 3.4. When Heart Rate Was Controlled, Ouabain Administration Reduced Stroke Volume in α_2_^+/G301R^ Mice

When heart rate was kept at 11.3 Hz (678 BPM) by atrial pacing, ouabain administration significantly increased both systolic (*p* = 0.0002) and diastolic blood pressure (*p* < 0.0001) in both α_2_^+/G301R^ and WT mice (Figure 6a,b). This increase in blood pressure was accompanied with an ouabain-induced augmentation of total peripheral resistance in both genotypes (*p* = 0.0001; Figure 6c).

A general effect of ouabain on stroke volume was detected during atrial pacing (*p* = 0.0054); however, following post-hoc analysis, ouabain administration reduced stroke volume during atrial pacing at 11.3 Hz in α_2_^+/G301R^ mice (*p* = 0.0473), whereas the change in stroke volume in WT did not achieve statistical significance (*p* = 0.1062; Figure 6d).

Arterial dP/dt_max_ and arterial dP/dt_min_ were unchanged after ouabain administration during atrial pacing in both groups (Figure 6e,f).

Blood pressure, total peripheral resistance, and stroke volume were unaffected by vehicle injection in both groups during atrial pacing at 11.3 Hz (Appendix A). However, when the hearts were paced at 11.3 Hz, arterial dP/dt_max_ was greater in WT mice compared to α_2_^+/G301R^ mice after vehicle administration (*p* = 0.0192; Appendix A). There was no change to arterial dP/dt_min_ (Appendix A).

### 3.5. When Heart Rate Was Controlled at 11.3 Hz, Ouabain-Induced Changes to Systolic Time Intervals in α_2_^+/G301R^ Mice Were Abolished

When the hearts were paced at 11.3 Hz, variables related to systolic time intervals including the systolic time interval ratio, pre-ejection period, left ventricular ejection time, and systolic duration were similar between genotypes before and after ouabain administration (Figure 7). Similarly, the systolic time interval ratio, pre-ejection period, left ventricular ejection time, and systolic duration were not affected by vehicle injection when the hearts were paced at 11.3 Hz (Appendix A).

## 4. Discussion

We aimed to characterize the Na^+^,K^+^-ATPase α_2_-isoform-specific contribution to cardiovascular function and, thus, investigated the comprehensive cardiovascular responses to ouabain of α_2_^+/G301R^ and WT mice. Additionally, we paced the hearts to discern potential rate-dependent changes to cardiac function associated to ouabain or the phenotype of the α_2_^+/G301R^ mouse model [48,49].

We hypothesized that the cardiovascular response to ouabain in α_2_^+/G301R^ mice was augmented due to their reduced expression of the Na^+^,K^+^-ATPase α_2_-isoform compared to WT. Pharmacologically, at a specific drug concentration, the effect of a drug may be described by how many receptors it binds relative to the total amount of receptors expressed on the cell membrane, i.e., fractional occupancy [50]. Mathematically, fractional occupancy equals the occupied binding sites divided by the number of total binding sites [50]. Thus, the fewer α_2_-isoforms present in the cell membrane of α_2_^+/G301R^ mice results in higher fractional occupancy and effect of a specific concentration of ouabain [9,11].

Accordingly, we found that the peripheral vascular effect of ouabain in vivo was augmented in the α_2_^+/G301R^ mice compared to WT. This was associated with ouabain-induced changes in the temporal development of the systole in α_2_^+/G301R^ mice. However, stroke volume remained unchanged during sinus rhythm in α_2_^+/G301R^ mice despite an increased total peripheral resistance and change in systolic function. Even so, stroke volume was reduced in α_2_^+/G301R^ mice during atrial pacing after ouabain administration. Thus, in this study we demonstrated an association between the ouabain-sensitive vascular phenotype in α_2_^+/G301R^ mice with reduced cardiac efficiency.

Different ouabain-sensitivity of the α_1_- and α_2_-isoforms of Na^+^,K^+^-ATPase in rodents [8] allows isoform-specific differentiation of their functions in the cardiovascular system [51]. The isoform-specific half-maximal inhibitory concentrations, i.e., IC50, of ouabain were reported to be 48,000 nM and 58 nM for the α_1_- and α_2_-isoforms in rats, respectively [28]. In our study, the basal plasma level of endogenous ouabain and its concentration after injection of exogenous ouabain were similar between α_2_^+/G301R^ and WT mice. We estimated that the basal concentration of ouabain in mouse plasma was approximately 300 pM. This concentration is in close agreement with previously reported concentrations for rodents and humans [52,53]. Furthermore, in this study, injection of exogenous ouabain elevated the plasma ouabain concentration to approximately 100 nM, i.e., a concentration that affects the α_2_-isoform and not the α_1_-isoform function. In our preliminary study, higher concentrations of ouabain were lethal presumably due to complete inhibition of the Na^+^,K^+^-ATPase emphasizing the vital importance of the Na^+^,K^+^-ATPase function for living cells. Furthermore, the Na^+^,K^+^-ATPase α_3_-isoform is mostly allocated to neuronal tissues [2], and the blood–brain barrier is not permeable to circulating ouabain [54]. Therefore, we anticipated that the cardiovascular α_2_-isoform was the primary target of the pharmacological intervention in this study. However, we cannot exclude that some of the effects of ouabain administration were mediated via the autonomic nervous system [55].

In our earlier study, we found that during nighttime, i.e., the active period of mouse circadian rhythm, aged (>8 months old) α_2_^+/G301R^ mice exhibited reduced blood pressure compared to WT [11]. However, blood pressure during daytime, i.e., the inactive period of mouse circadian rhythm, was the same between the two genotypes [11]. Accordingly, no difference in blood pressure between genotypes was seen in this study in anesthetized mice under baseline conditions.

In accordance with previous reports on rodents, we found that ouabain administration increased blood pressure [29,56]. The ouabain-induced blood pressure elevation has previously been ascribed to the inotropic effect of ouabain [31,32]. Paradoxically, ouabain was also reported to potentiate not only vasoconstriction but also vasodilation [9,57,58,59]. In our study, ouabain-induced elevation of blood pressure was associated with augmentation of total peripheral resistance, which was most pronounced in α_2_^+/G301R^ mice. The major contribution of total peripheral resistance in this ouabain-induced blood pressure elevation was further supported by unaltered cardiac functions in both groups after ouabain administration. In line with previous ex vivo reports [9,57,60,61], our results suggest a primary pro-contractile action of ouabain on the vasculature at concentrations specific for α_2_-isoform inhibition. We propose that reduction in the vascular expression of the α_2_-isoform underlies the augmented ouabain-induced increase in total peripheral resistance in α_2_^+/G301R^ mice. Accordingly, a similar concentration of circulating ouabain has a more efficient inhibitory action in α_2_^+/G301R^ mice than in WT controls. Hence, our findings add to the perception of the importance of the α_2_-isoform function for vascular tone and blood pressure in vivo in rodents as previously proposed [62,63].

Systolic time intervals entail information about the overall external myocardial efficiency [43]. The systolic time interval ratio increased only in α_2_^+/G301R^ mice after ouabain administration suggesting a decrease in myocardial efficiency. This increase in the systolic time interval ratio was a result of both a significant extension of the pre-ejection period and a slight shortening of the left ventricular ejection time. We suggest that the prolonged pre-ejection period may be a result of prolongation of the isovolumetric contraction phase of the heart as a result of the augmented ouabain-induced increase in total peripheral resistance in α_2_^+/G301R^ mice [44]. However, we cannot exclude that ouabain induced changes in the electrical properties of the heart affecting the pre-ejection period [64], albeit heart rate was unchanged.

The inotropic effect of ouabain is well characterized, and it is described as a cardiac calcitrope, i.e., the inotropic effect is mediated by an increase in intracellular Ca^2+^ [27,65]. In our study, we assessed the arterial dP/dt_max_ as a proxy for cardiac contractility [41]. However, unlike the dP/dt_max_ measured directly in the left ventricle [66], the arterial dP/dt_max_ also depends on arterial system properties, e.g., the stiffness of the arterial wall [41]. In the current study, the arterial dP/dt_max_ was not affected following ouabain or vehicle administration during sinus rhythm. Considering that stroke volume and cardiac output did not change despite augmentation of total peripheral resistance in both WT and α_2_^+/G301R^ mice, the presence of an inotropic effect of ouabain in our study during sinus rhythm is plausible. Additionally, shortening of left ventricular ejection time has been associated with an increase in intracellular Ca^2+^ mediated by cardiac calcitropes [43]. As left ventricular ejection time was only shortened in α_2_^+/G301R^ mice, our finding may suggest a stronger cardiac effect of ouabain in α_2_^+/G301R^ mice during sinus rhythm. In accordance, stroke volume and cardiac output were maintained in α_2_^+/G301R^ mice despite greater augmentation of total peripheral resistance during sinus rhythm.

During atrial pacing, the arterial dP/dt_max_ was lower in α_2_^+/G301R^ mice compared to WT after vehicle administration indicating a deleterious effect of time on the contractility in α_2_^+/G301R^ mice. This was rescued by ouabain administration. However, the ouabain-induced recovery of arterial dP/dt_max_ in α_2_^+/G301R^ mice compared to WT may also be a consequence of the augmented total peripheral resistance in α_2_^+/G301R^ mice. In addition, the changes to systolic time intervals in α_2_^+/G301R^ mice were not seen during atrial pacing. The systolic time intervals have been described to be influenced by heart rate, especially, in conjunction with hemodynamic changes [67,68,69]. Thus, in the current study, we suggest that the high heart rates (678 BPM) during atrial pacing compared to the heart rates during sinus rhythm (approximately 532 BPM) limits deviation of the systolic time intervals. This was further associated with reduced stroke volume in α_2_^+/G301R^ mice. Thus, our findings may also suggest rate-dependent cardiac effects of ouabain.

Our study proposes a detrimental link between the vascular phenotype in α_2_^+/G301R^ mice [9,10] and their cardiac function [11]. We suggest that the changes to cardiac function in α_2_^+/G301R^ mice described in this study reflect the reduced ventricular function previously reported for aged α_2_^+/G301R^ mice [11]. Furthermore, we suggest that the changed vascular reactivity, possibly in combination with intrinsic cardiac changes [11], underlies the pathological cardiovascular changes seen in α_2_^+/G301R^ mice. This is based on the following observations from this and previous studies [9,10,11]: (i) the changed cardiac phenotype in α_2_^+/G301R^ mice develops over time and is only seen in aged mice, (ii) the vascular phenotype in α_2_^+/G301R^ mice is not unique to a specific age, and (iii) the presence of reduced cardiac efficiency in α_2_^+/G301R^ mice when challenged with increments in afterload demonstrated in this study. However, whether parts of the cardiac changes are pre-defined or solely a product of the vascular abnormalities remain to be clarified. Nevertheless, all these aspects of cardiovascular dysfunction in α_2_^+/G301R^ mice may in concert facilitate a milieu that leads to cardiac pathology over time [70]. Often, this type of cardiac disease progression is slow and silent, and it may resemble a heart failure phenotype with no apparent clinical symptoms during rest and sinus rhythm [71]. Similarly, the cardiac phenotype in α_2_^+/G301R^ mice was only revealed during atrial pacing or after ouabain administration, and there was no apparent phenotype during sinus rhythm and at baseline in this study.

We primarily identified the changed cardiovascular phenotype in α_2_^+/G301R^ mice. Intriguingly, the α_2_^+/G301R^ mouse is also an animal model for migraine [21]. Migraine has been described as a significant risk factor in the development of cardiovascular morbidity [72,73]. However, the underlying mechanism remains unclear. Studies have suggested that migraine may be considered a “channelopathy”, specifically, that defects in ion transporters are the primary culprits in migraine pathology [74,75]. Furthermore, studies have argued that migraine should be considered a systemic disease rather than an isolated neurovascular condition [76,77]. Accordingly, our data suggest that global reduction of cardiovascular Na^+^,K^+^-ATPase α_2_-isoform associated with the migraine-associated mutation changes the vascular phenotype with consequences for cardiac function in α_2_^+/G301R^ mice. This notion warrants investigation of the cardiovascular phenotype of FHM2 migraineurs, which may further elucidate the molecular link between cardiovascular morbidity and migraine.

## 5. Conclusions

Our results indicate that both the cardiac and vascular phenotypes are affected in α_2_^+/G301R^ mice in vivo. Our findings indicate less resilience of α_2_^+/G301R^ hearts to the Na^+^,K^+^-ATPase α_2_-isoform inhibition with ouabain. Specifically, α_2_^+/G301R^ mice showed an augmented increase in total peripheral resistance associated with decreased cardiac efficiency in the presence of ouabain. We suggest that the hypercontractile vascular phenotype of the α_2_^+/G301R^ mice may lead to detrimental cardiac function over time.

## Figures and Tables

**Figure 1 biomedicines-11-00344-f001:**
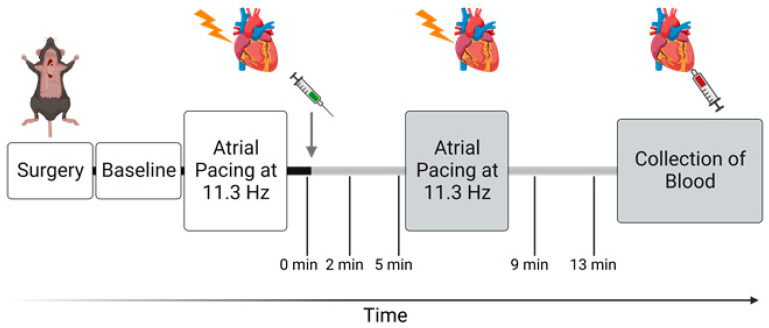
**Timeline of the in vivo protocol**. Gray arrow indicates point of intraperitoneal injection of either ouabain (1.5 mg/kg) or vehicle (0.9 mg/mL NaCl), and gray bars indicate period after administration. After implantation of probes, a baseline measurement was made. The protocol includes atrial pacing starting from 10 Hz (600 BPM) with stepwise increments to 11.3 Hz (678 BPM) before and after administration. Time points 0, 2, 5, 9, and 13 min represent measurements done during sinus rhythm. The protocol was finalized by blood collection from the apex 15 min after administration of ouabain or saline. Figure was created with BioRender.com.

**Figure 2 biomedicines-11-00344-f002:**
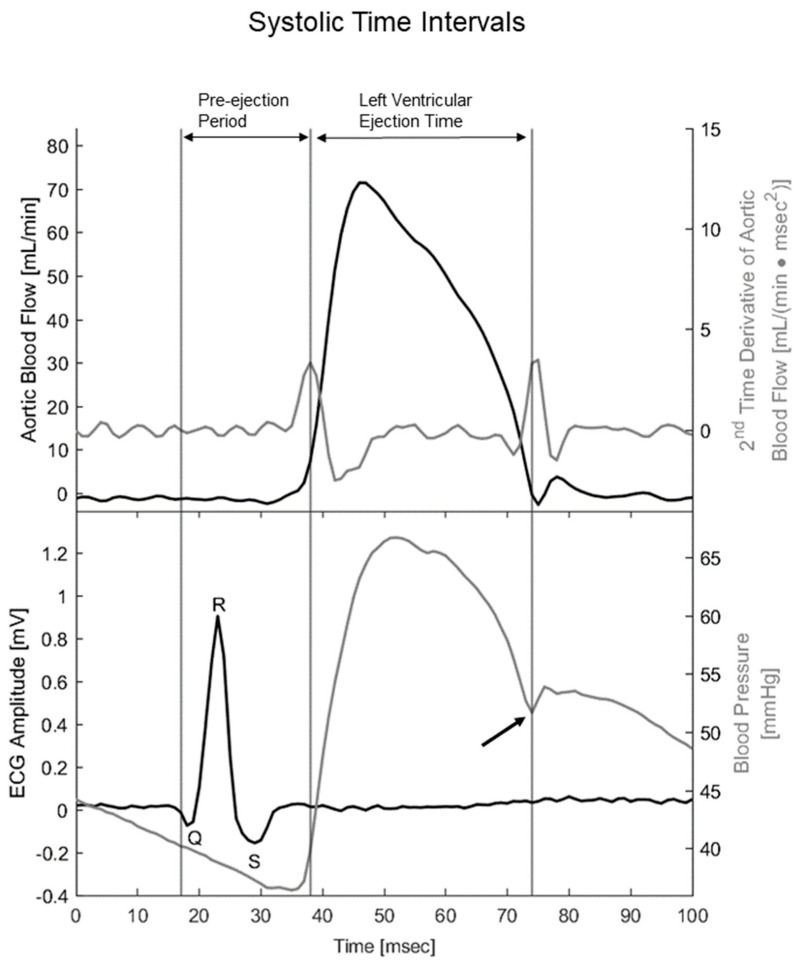
Temporal alignment between the second time derivative of aortic blood flow, ECG, and blood pressure traces allowed for the calculation of systolic time intervals. Upper panel includes blood flow through the ascending aorta (black line) and its 2nd time derivative, i.e., acceleration (gray line). Lower panel includes mouse ECG lead I with depicted Q, R, and S waves (black line) together with arterial blood pressure measured in the aortic arc (gray line). Black arrow indicates the dicrotic notch on the blood pressure trace, i.e., closure of the aortic valve. Preejection period was defined as spanning from the beginning of the Q wave to opening of the aortic valve, i.e., Q wave to the first peak of the second time derivative of aortic flow. Left ventricular ejection period was defined as spanning from the opening to the closure of the aortic valve, i.e., spanning from the first peak of the second time derivative of blood flow to the dicrotic notch. Systolic duration was defined as the sum of the pre-ejection period and left ventricular ejection time.

**Figure 3 biomedicines-11-00344-f003:**
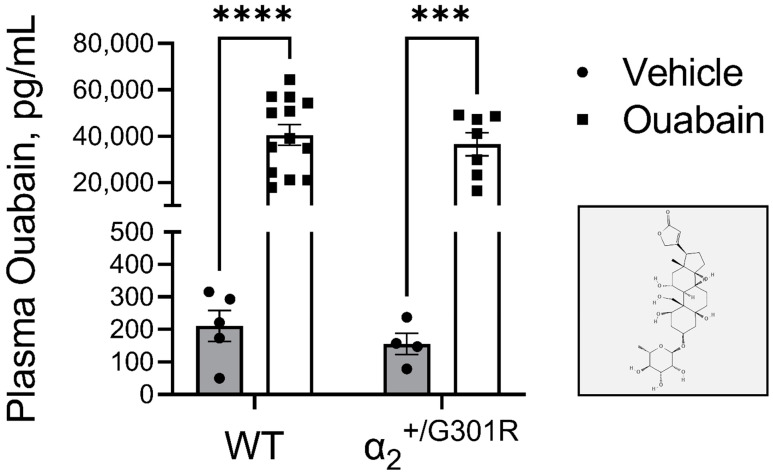
Similar plasma ouabain concentrations between wild-type (WT) and α_2_^+/G301R^ mice approximately 15 mins after injection of ouabain (1.5 mg/kg) or vehicle (0.9% NaCl) treatment. Injection of ouabain i.p. increased plasma ouabain concentrations similarly in both WT (210 ± 48 pg/mL and 40,517 ± 4438 pg/mL for vehicle and ouabain administered mice, respectively) and α_2_^+/G301R^ (156 ± 33 pg/mL and 36,489 ± 5041 pg/mL for vehicle and ouabain administered mice, respectively) mice compared to their respective vehicle groups. (Insert) The molecular structure of ouabain in 2D (adopted from the PubChem database [47]). ELISA data were analyzed by two-way ANOVA followed by Bonferroni’s multiple comparison test. *** *p* = 0.0003 and **** *p* < 0.0001; intragroup plasma ouabain concentration of vehicle vs. ouabain treated mice. *n* = 7–13 for ouabain groups, and *n* = 4–5 for vehicle groups.

**Figure 4 biomedicines-11-00344-f004:**
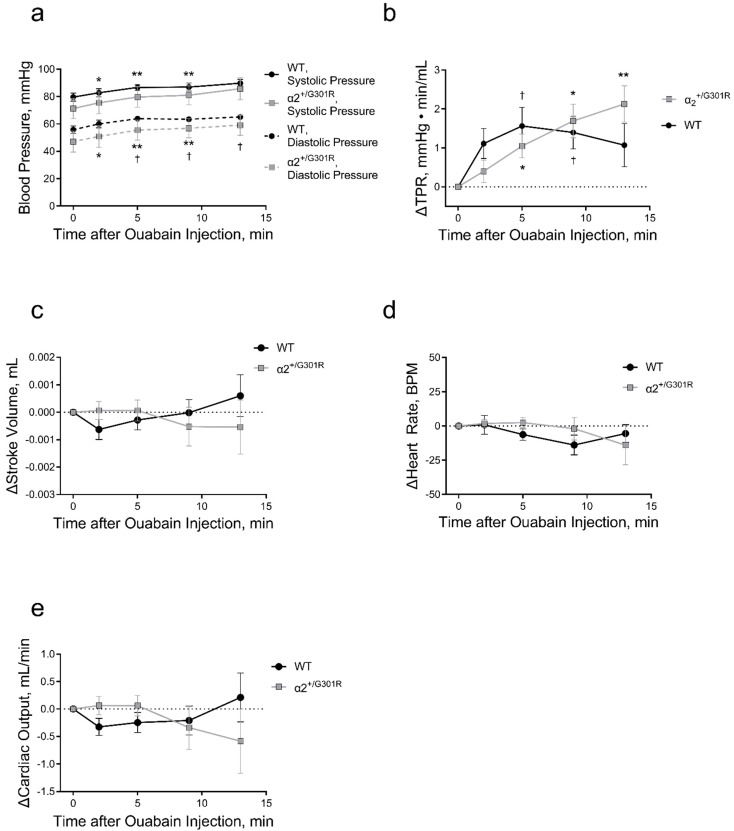
Intraperitoneal ouabain injection (1.5 mg/kg) increased blood pressure and total peripheral resistance in wild-type (WT) and α_2_^+/G301R^ mice. (**a**) Ouabain increased blood pressure in WT and α_2_^+/G301R^ mice (systolic pressure; *p* = 0.0193, and diastolic pressure; *p* = 0.0100). (**b**) Ouabain-induced increase in total peripheral resistance (TPR) was different between α_2_^+/G301R^ and WT mice (*p* = 0.0340). There was no significant effect of ouabain administration on stroke volume (**c**), heart rate (**d**), or cardiac output (**e**) in neither WT nor α_2_^+/G301R^ mice. Data were compared with two-way repeated measures ANOVA followed by Dunnett’s multiple comparison test. * *p* < 0.05 and ** *p* < 0.01 for α_2_^+/G301R^ and † *p* < 0.05 for WT; variables at different time points vs. intragroup baseline before ouabain administration, i.e., time 0 min. *n* = 8. Data are available in Appendix A.

**Figure 5 biomedicines-11-00344-f005:**
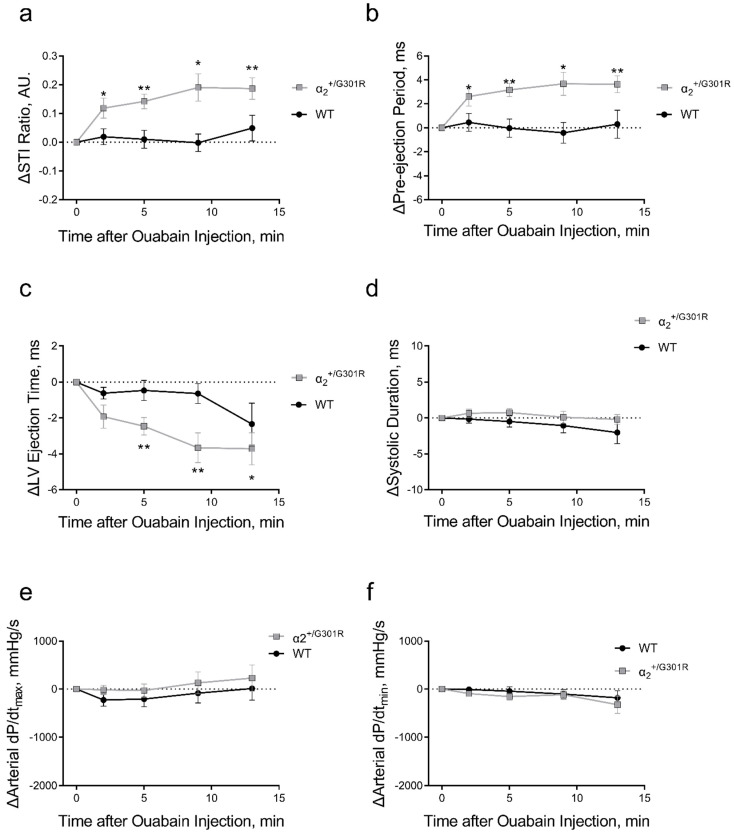
Systolic time intervals were affected by ouabain administration (1.5 mg/kg, i.p.) only in α_2_^+/G301R^ mice and not in wild-type mice (WT). (**a**) the systolic time interval ratio (STI Ratio) was significantly increased in α_2_^+/G301R^ mice only (*p* = 0.0004) due to (**b**) prolongation of pre-ejection period (*p* = 0.0007) and (**c**) shortening of left ventricular ejection time (LV Ejection Time; *p* = 0.0426). (**d**) Systolic duration, (**e**) arterial dP/dt_max_, and (**f**) arterial dP/dt_min_ were unchanged in both WT and α_2_^+/G301R^ mice after ouabain administration. Data were compared with two-way repeated measures ANOVA followed by Dunnett’s multiple comparison test. * *p* < 0.05 and ** *p* < 0.01 for α_2_^+/G301R^; variables at different time points vs. intragroup baseline before ouabain administration, i.e., time 0 min. *n* = 8. Data are available in Appendix A.

**Figure 6 biomedicines-11-00344-f006:**
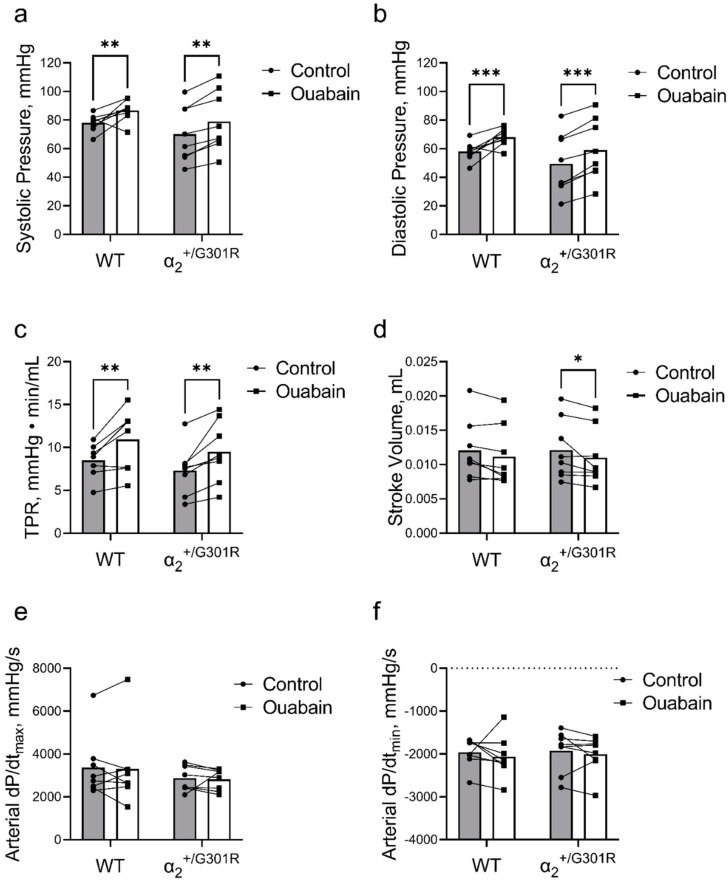
Ouabain administration increased blood pressure and total peripheral resistance in both wild-type (WT) and α_2_^+/G301R^ mice when the hearts were paced at 11.3 Hz (678 BPM). Under these conditions, ouabain reduced stroke volume in α_2_^+/G301R^ mice. Administration of ouabain augmented (**a**) systolic blood pressure (control vs. ouabain; 78.01 ± 2.14 mmHg vs. 86.71 ± 2.67 mmHg for WT and 70.19 ± 6.90 mmHg vs. 78.94 ± 7.50 mmHg for α_2_^+/G301R^ mice; *p* = 0.0002), (**b**) diastolic blood pressure (control vs. ouabain; 58.01 ± 2.31 mmHg vs. 68.01 ± 2.16 mmHg for WT and 49.50 ± 7.51 mmHg vs. 59.03 ± 7.56 mmHg for α_2_^+/G301R^ mice; *p* < 0.0001), and (**c**) total peripheral resistance (TPR; control vs. ouabain; 8.49 ± 0.68 mmHg·min/mL vs. 10.92 ± 1.24 mmHg·min/mL for WT and 7.31 ± 1.00 mmHg·min/mL vs. 9.48 ± 1.25 mmHg·min/mL for α_2_^+/G301R^ mice; *p* = 0.0001). Ouabain administration significantly decreased stroke volume (**d**) in α_2_^+/G301R^ mice (control vs. ouabain; 12.12 ± 1.54 µL vs. 11.02 ± 1.45 µL; *p* = 0.0473), whereas changes to stroke volume in WT mice did not achieve statistical significance (control vs. ouabain; 12.07 ± 1.53 µL vs. 11.15 ± 1.54 µL; *p* = 0.1062). (**e**) Arterial dP/dt_max_ (control vs. ouabain; 3367 ± 516 mmHg/s vs. 3303 ± 629 mmHg/s for WT and 2872 ± 209 mmHg/s vs. 2820 ± 174 mmHg/s for α_2_^+/G301R^ mice) and (**f**) arterial dP/dt_min_ (control vs. ouabain; −1958 ± 119 mmHg/s vs. −2067 ± 172 mmHg/s for WT and −1922 ± 172 mmHg/s vs. −2009 ± 155 mmHg/s for α_2_^+/G301R^ mice) were unchanged in both groups. Data were compared with multiple Wilcoxon tests or two-way repeated measures ANOVA followed by Bonferroni’s multiple comparison test. * *p* < 0.05, ** *p* < 0.01, and *** *p*< 0.001; intragroup variables before vs. after ouabain administration. *n* = 8.

**Figure 7 biomedicines-11-00344-f007:**
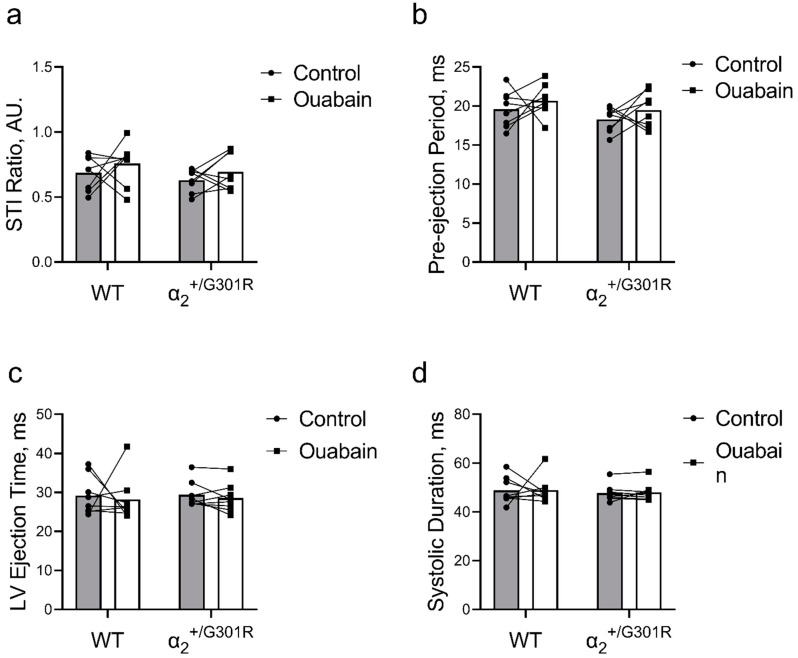
Ouabain administration was without an effect on systolic time intervals in α_2_^+/G301R^ and wild-type mice (WT) when the hearts were paced at 11.3 Hz (678 BPM). (**a**) The systolic time interval ratio (STI Ratio; control vs. ouabain; 0.69 ± 0.04 vs. 0.76 ± 0.06 for WT and 0.63 ± 0.03 vs. 0.69 ± 0.05 for α_2_^+/G301R^ mice), (**b**) pre-ejection period (control vs. ouabain; 19.6 ± 0.8 ms vs. 20.7 ± 0.7 ms for WT and 18.3 ± 0.5 ms vs. 19.5 ± 0.8 ms for α_2_^+/G301R^ mice), (**c**) left ventricular ejection time (LV ejection time; control vs. ouabain; 29.2 ± 1.8 ms vs. 28.2 ± 2.1 ms for WT and 29.4 ± 1.2 ms vs. 28.6 ± 1.3 ms for α_2_^+/G301R^ mice), and (**d**) systolic duration (control vs. ouabain; 48.8 ± 2.0 ms vs. 49.0 ± 1.9 ms for WT and 47.7 ± 1.2 ms vs. 48.1 ± 1.3 ms for α_2_^+/G301R^ mice) were similar during atrial pacing at 11.3 Hz before and after ouabain administration. Data were compared with two-way repeated measures ANOVA. *n* = 8.

**Table 1 biomedicines-11-00344-t001:** Baseline in vivo parameters and cardiovascular variables of all wild-type and α_2_^+/G301R^ mice were similar. One wild-type mouse injected with vehicle had an uninterpretable ECG recording and, thus, was excluded in systolic time interval calculations. Data were compared with an unpaired *t*-test or a Mann–Whitney test where appropriate.

	Wild Type (n = 12)	α_2_^+/G301R^ (n = 12)	*p*-Value
Age (weeks)	41 ± 2	45 ± 3	0.23
Body mass (g)	35.3 ± 1.6	35.3 ± 0.8	>0.99
**Cardiovascular variables and respiratory parameters**			
Systolic pressure (mmHg)	76.09 ± 2.97	72.09 ± 4.68	0.48
Diastolic pressure (mmHg)	49.71 ± 3.53	46.16 ± 5.00	0.57
Total peripheral resistance (mmHg•min/mL)	6.89 ± 0.71	7.21 ± 0.73	0.76
Cardiac output (mL/min)	9.13 ± 0.75	8.28 ± 0.93	0.48
Stroke volume (µL)	17.20 ± 1.50	15.66 ± 1.83	0.52
Heart rate (BPM)	534 ± 8	530 ± 10	0.81
End-tidal CO_2_ (%)	3.69 ± 0.13	3.76 ± 0.11	0.69
Ventilation rate (min^−1^)	116 ± 6	114 ± 7	0.87
Systolic time interval ratio (AU.)	0.59 ± 0.04	0.54 ± 0.03	0.27
Pre-ejection period (ms)	19.2 ± 0.8	18.2 ± 0.5	0.28
Left ventricular ejection time (ms)	33.2 ± 1.5	34.4 ± 1.6	0.58
Systolic duration (ms)	52.4 ± 1.6	52.6 ± 1.8	0.82
Arterial dP/dt_max_ (mmHg/s)	3705 ± 298.9	3142 ± 197.9	0.13
Arterial dP/dt_min_ (mmHg/s)	−2157 ± 92.78	−2106 ± 134.4	0.76

## Data Availability

The datasets generated and analyzed during the current study are available from the corresponding author on reasonable request.

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
