# Peer review of "Augmented Ouabain-Induced Vascular Response Reduces Cardiac Efficiency in Mice with Migraine-Associated Mutation in the Na+, K+-ATPase α2-Isoform"

_biomedicines, 2023, doi:10.3390/biomedicines11020344_

Round 1
Reviewer 1 Report
The manuscript of Rajanthan et al. entitled: „Augmented quabain-induced vascular response reduces cardiac efficiency in mice with migraine-associated mutation in the Na+,K+-ATPase α2-isoform“ deals with the functional effect of the migraine-associated G301R mutation in the gene encoding the Na+,K+-ATPase α2-isoform on the cardiac function. Quabain, a cardiac glycoside, was employed as a probe to characterize the functional contribution of the Na+/K+-ATPase to cardiac function/dysfuntion. A comprehensive cardiovascular response to quabain was studies in heterozygous mice carrying G301R mutation (α2+/G301R) and WT mice. Without atrial pacing and pharmacological intervention, any difference was detected between two genotypes, testing physiological variables related to the cardiac function. Quabain administration without atrial pacing increased blood pressure (systolic/diastolic) and the total peripheral resistance in both groups. However, only in mutant mice, quabain prolonged pre-ejection period and shortened the left ventricular ejection time. In addition, when the heart rate was controlled by atrial pacing at 11.3 Hz, quabain reduced stroke volume only in mutant mice. Other parameters related to the systolic duration were not affected by quabain when atrial pacing was involved. From such observations, the authors made a final conclusion that both the cardiac and vascular phenotypes are affected in mutant mice α2+/G301R in vivo.
Reading the manuscript and appreciating the arguments, I identified the following major and minor issues, which need to be addressed by the authors.
Major points
(1.) Introduction: I recommend to add a short section about the physiological role of the Na+/K+-ATPase in cardiomyocytes.
(2.) Results: The authors concluded that blood pressure in both groups was increased after administration of quabain (results shown in Figure 1). However, statistical evaluation for WT systolic pressure is missing.
(3.) Results: Results should be described in more detail. Now only, outcomes of statistical evaluation are mentioned (P values). The average value ± SEM should be also mentioned, because it was quite difficult for me to get such information only from graphs.
(4.) Results: What was the reason to change the format for data presentation in Figure 6 and 7? Similar data were shown in previous Figures displaying only average values ± SEM.
(5.) Results: Please explain, why the augmented effect of quabain on pre-ejection period and ejection time in mutant mice was not observed in case the heart rate was controlled by atrial pacing. What was the reason to use atrial pacing?
(6.) Results: It was mentioned that reduction of stroke volume during atrial pacing was observed only for mutant mice and not for WT (Figure 6d). However, presented significance for mutant mice is not so strong (p=0.0473). After a careful inspection of experimental data, it seems that only one larger decrease contributed to this significance. Other values could be comparable to those of WT. Adding more data for better characterization of the experimental groups could help to make a more appropriate conclusion regarding the decrease in stroke volume.
(7.) Discussion: The authors proposed that the augmented effect of quabain on periperal resistance in mutant mice is related to the suppressed expression of the mutated Na+/K+-ATPase α2-isoform. I would not make such a strong conclusion without testing sensitivity of the mutated α2-isoform to quabain. Is it known how G301R mutation affects the functional profile of the Na+/K+-ATPase α2-isoform? It is feasible to assume that increased sensitivity of the mutant protein to quabain could also contribute to changes between tested genotypes.
Minor points
(1.) Graphical design of the manuscript should be definitely improved. The big gap on the page 8 appeared that has to be removed. I suggest decreasing size of all figures and using a more compact form.
(2.) Figure 6: Mutation inserted in the gene is G301R not only 301R.
Author Response
We thank the reviewer for the comments. We believe they have improved the manuscript and increased its readability.
Reviewer 1
The manuscript of Rajanthan et al. entitled: „Augmented quabain-induced vascular response reduces cardiac efficiency in mice with migraine-associated mutation in the Na+,K+-ATPase α2-isoform“ deals with the functional effect of the migraine-associated G301R mutation in the gene encoding the Na+,K+-ATPase α2-isoform on the cardiac function. Quabain, a cardiac glycoside, was employed as a probe to characterize the functional contribution of the Na+/K+-ATPase to cardiac function/dysfuntion. A comprehensive cardiovascular response to quabain was studies in heterozygous mice carrying G301R mutation (α2+/G301R) and WT mice. Without atrial pacing and pharmacological intervention, any difference was detected between two genotypes, testing physiological variables related to the cardiac function. Quabain administration without atrial pacing increased blood pressure (systolic/diastolic) and the total peripheral resistance in both groups. However, only in mutant mice, quabain prolonged pre-ejection period and shortened the left ventricular ejection time. In addition, when the heart rate was controlled by atrial pacing at 11.3 Hz, quabain reduced stroke volume only in mutant mice. Other parameters related to the systolic duration were not affected by quabain when atrial pacing was involved. From such observations, the authors made a final conclusion that both the cardiac and vascular phenotypes are affected in mutant mice α2+/G301R in vivo.
Reading the manuscript and appreciating the arguments, I identified the following major and minor issues, which need to be addressed by the authors.
Major points
- Introduction: I recommend to add a short section about the physiological role of the Na+/K+-ATPase in cardiomyocytes.
A brief introduction to the functions of the Na+,K+-ATPase together with its importance for NCX function and Ca2+ regulation has been provided in the introduction. See line 31-57.
(2.) Results: The authors concluded that blood pressure in both groups was increased after administration of quabain (results shown in Figure 1). However, statistical evaluation for WT systolic pressure is missing.
In general, there was an effect of ouabain on blood pressure in both genotypes. However, following post-hoc analysis (Dunnett’s multiple comparison), the values for systolic pressure in the WT mice did not achieve statistical significance (P = 0.09, P = 0.09, P = 0.056, P = 0.09 for x = 2, 5, 9, and 13 minutes, respectively, when compared to the time prior ouabain administration). This is now made clear in the text; see lines 273-275.
(3.) Results: Results should be described in more detail. Now only, outcomes of statistical evaluation are mentioned (P values). The average value ± SEM should be also mentioned, because it was quite difficult for me to get such information only from graphs.
We have edited the manuscript accordingly. To not encumber some readers too much, we have added mean ± SEM for measurements over time in Supplementary tables 1 and 2 (see Supplementary material). The mean values ± SEM for pacing data are now added to their respective figure legends. We believe that this is a nice compromise between the readability of the main text and access to more detailed information about the results in the legends and tables.
(4.) Results: What was the reason to change the format for data presentation in Figure 6 and 7? Similar data were shown in previous Figures displaying only average values ± SEM.
In Figure 4 and 5, we emphasized the temporal development in cardiovascular variables in response to intraperitoneal ouabain injection, hence, we found the current format useful in conveying this message. Due to the many time points in the graph (0, 2, 5, 9, and 13 minutes), displaying each individual point together with their connecting lines will muddy the graphs and obscure their messages. In the pacing data, however, we wanted to indicate the paired data before and after the administration of the vehicle or ouabain. Thus, we decided to display the pairing of individual values in the current format. We believe that with the addition of the mean ± SEM in the figure legends, the current format conveys a clear message of the changes in cardiovascular variables during sinus rhythm, atrial pacing, before and after intraperitoneal injection of ouabain and vehicle.
(5.) Results: Please explain, why the augmented effect of quabain on pre-ejection period and ejection time in mutant mice was not observed in case the heart rate was controlled by atrial pacing. What was the reason to use atrial pacing?
This has now been addressed in the discussion. We explained this by the rate-dependence of systolic time intervals (see line 464-469). We conducted atrial pacing to discern potential rate-dependent effects of ouabain and/or the cardiovascular phenotype of the α2+/G301R mouse model (see line 375-377).
(6.) Results: It was mentioned that reduction of stroke volume during atrial pacing was observed only for mutant mice and not for WT (Figure 6d). However, presented significance for mutant mice is not so strong (p=0.0473). After a careful inspection of experimental data, it seems that only one larger decrease contributed to this significance. Other values could be comparable to those of WT. Adding more data for better characterization of the experimental groups could help to make a more appropriate conclusion regarding the decrease in stroke volume.
We agree that it is a subtle decrease in stroke volume in α2+/G301R mice, however, we do also suggest in the last paragraph of the discussion section that it is exactly subtle cardiac changes over time that facilitate the cardiac phenotype in the α2+/G301R mouse model. A general effect of ouabain on stroke volume was found in the two-way repeated measures ANOVA analysis (P = 0054), and this has now been made clear in the results section (see line 349-352). Furthermore, we have removed “only in α2+/G301R mice” throughout the text.
(7.) Discussion: The authors proposed that the augmented effect of quabain on periperal resistance in mutant mice is related to the suppressed expression of the mutated Na+/K+-ATPase α2-isoform. I would not make such a strong conclusion without testing sensitivity of the mutated α2-isoform to quabain. Is it known how G301R mutation affects the functional profile of the Na+/K+-ATPase α2-isoform? It is feasible to assume that increased sensitivity of the mutant protein to quabain could also contribute to changes between tested genotypes.
We agree with the reviewer that we cannot fully exclude this possibility. However, a previous study characterized the G301R mutation in cell culture and reported that the G301R mutation impedes translocation of the α2-isoform to the cell membrane and in this way prevents the mutated protein to be functional (DOI: 10.1177/0333102411399351). Accordingly, we have previously shown that the expression of α2-isoform on both vascular smooth muscle cell and endothelial cells is reduced by approximately 50% (line 65 of the manuscript).
To elaborate our rationale for the hypothesis that ouabain has a greater effect in the α2+/G301R mouse model due to fewer α2-isoform proteins in the membrane, we added a section clarifying this issue in the discussion (see line 378-385).
Minor points
- Graphical design of the manuscript should be definitely improved. The big gap on the page 8 appeared that has to be removed. I suggest decreasing size of all figures and using a more compact form.
We do not expect that this will be an issue in the published version after publishing office adjustments. We believe that further decreasing the size of figures in the current version for the review will reduce readability of the manuscript.
- Figure 6: Mutation inserted in the gene is G301R not only 301R.
This has now been corrected, thank you. Furthermore, similar mistakes were found in Figure 7, Supplementary Figures 3, and 4. They have also been corrected now.
Reviewer 2 Report
Oubain (g-strophanthin) was found in the mature seeds of the pleasant strophanthus (Strophanthus gratus) and in the bark of the Abyssinian acokanthera (Acokanthera schimperi) (syn. Acokanthera ouabaio). Both plants are native to Africa. Ouabain is known to be a cardiac glycoside and in lower doses, can be used medically to treat hypotension and some arrhythmias. The classical mechanism of action of ouabain includes binding and inhibition of the action of the membrane protein of the sodium-potassium pump: Na+/K+-ATPase in the eukaryotic cell.
The team of authors has done a wonderful and voluminous work, which is undoubtedly of great scientific and practical interest, which allows us to recommend this article for publication.
Meanwhile, I have a number of comments and suggestions that should be taken into account when preparing the manuscript for publication:
1. I think that it would not be superfluous to give the chemical structure of ouabain, since this article can also be read by specialists in the field of medicinal chemistry and the chemistry of natural compounds.
2. Previously, the properties of ouabain were studied by a large number of researchers, for example, J Cardiovasc Pharmacol. 2013 Aug;62(2):174-83. doi: 10.1097/FJC.0b013e3182955d33; Br J Pharmacol. 2004 Sep;143(1):215-25. doi: 10.1038/sj.bjp.0705919; J Vasc Res 2011;48:316–326. https://doi.org/10.1159/000322576; Proc Nat Acad Sci. 109(32): 13040–13045. doi:10.1073/pnas.1202111109 and etc, therefore, in the introductory part of the article, it is necessary to discuss in more detail the literature data and the knowledge achieved so far about the biological activity of ouabain and the mechanisms of its action. In conclusion, it is also necessary to speak more clearly about the future prospects for the practical use of ouabain and the discrepancies found with the literature data on the mechanism of its action.
Author Response
We thank the reviewer for their comments and suggestion. We have revised the manuscript accordingly and believe it has improved. Please, see our rebuttal below.
Reviewer 2
Oubain (g-strophanthin) was found in the mature seeds of the pleasant strophanthus (Strophanthus gratus) and in the bark of the Abyssinian acokanthera (Acokanthera schimperi) (syn. Acokanthera ouabaio). Both plants are native to Africa. Ouabain is known to be a cardiac glycoside and in lower doses, can be used medically to treat hypotension and some arrhythmias. The classical mechanism of action of ouabain includes binding and inhibition of the action of the membrane protein of the sodium-potassium pump: Na+/K+-ATPase in the eukaryotic cell.
The team of authors has done a wonderful and voluminous work, which is undoubtedly of great scientific and practical interest, which allows us to recommend this article for publication.
Meanwhile, I have a number of comments and suggestions that should be taken into account when preparing the manuscript for publication:
- I think that it would not be superfluous to give the chemical structure of ouabain, since this article can also be read by specialists in the field of medicinal chemistry and the chemistry of natural compounds.
We have added a 2D model of the molecular structure of ouabain in Figure 3.
- Previously, the properties of ouabain were studied by a large number of researchers, for example, J Cardiovasc Pharmacol. 2013 Aug;62(2):174-83. doi: 10.1097/FJC.0b013e3182955d33; Br J Pharmacol. 2004 Sep;143(1):215-25. doi: 10.1038/sj.bjp.0705919; J Vasc Res 2011;48:316–326. https://doi.org/10.1159/000322576; Proc Nat Acad Sci. 109(32): 13040–13045. doi:10.1073/pnas.1202111109 and etc, therefore, in the introductory part of the article, it is necessary to discuss in more detail the literature data and the knowledge achieved so far about the biological activity of ouabain and the mechanisms of its action. In conclusion, it is also necessary to speak more clearly about the future prospects for the practical use of ouabain and the discrepancies found with the literature data on the mechanism of its action.
We accommodated this suggestion of the reviewer with the following changes:
- In the introduction section, more background information has been added regarding ouabain’s molecular character and its function (see line 71-98).
- In the discussion section, we discussed that the vasoactive effect of ouabain may be attributed to the α2-isoform. This suggests that isoform-specific functions should be considered in investigations of ouabain effects in the cardiovascular system. This isoform-specificity has been emphasized and implied in both the introduction and the discussion sections regarding ouabain and its effect on blood pressure (also, see lines 418-433).